# Mutation in *ROBO*3 Gene in Patients with Horizontal Gaze Palsy with Progressive Scoliosis Syndrome: A Systematic Review

**DOI:** 10.3390/ijerph17124467

**Published:** 2020-06-22

**Authors:** Elena Pinero-Pinto, Verónica Pérez-Cabezas, Cristina Tous-Rivera, José-María Sánchez-González, Carmen Ruiz-Molinero, José-Jesús Jiménez-Rejano, María-Luisa Benítez-Lugo, María Carmen Sánchez-González

**Affiliations:** 1Department of Physiotherapy, University of Seville, 41009 Seville, Spain; epinero@us.es (E.P.-P.); jjjimenez@us.es (J.-J.J.-R.); marisabeni@us.es (M.-L.B.-L.); 2Department of Nursing and Physiotherapy, Spain INDESS (Instituto Universitario para el Desarrollo Social Sostenible), University of Cadiz, 11009 Cadiz, Spain; carmen.ruizmolinero@uca.es; 3Nodo Biobanco Hospital Universitario Virgen del Rocío (Biobanco del Sistema Sanitario Público de Andalucía), 41013 Seville, Spain; ctous@us.es; 4Department of Physics of Condensed Matter, Optics Area, University of Seville, 41012 Seville, Spain; jsanchez80@us.es (J.-M.S.-G.); msanchez77@us.es (M.C.S.-G.)

**Keywords:** mutation, gaze palsy, familial horizontal, scoliosis, children

## Abstract

Horizontal gaze palsy with progressive scoliosis (HGPPS) is a rare, inherited disorder characterized by a congenital absence of conjugate horizontal eye movements with progressive scoliosis developing in childhood and adolescence. Mutations in the Roundabout (*ROBO*3) gene located on chromosome 11q23–25 are responsible for the development of horizontal gaze palsy and progressive scoliosis. However, some studies redefined the locus responsible for this pathology to a 9-cM region. This study carried out a systematic review in which 25 documents were analyzed, following Preferred Reporting Items for Systematic Reviews and Meta-Analyses (PRISMA) standards. The search was made in the following electronic databases from January 1995 to October 2019: PubMed, Scopus, Web of Science, PEDRO, SPORT Discus, and CINAHL. HGPPS requires a multidisciplinary diagnostic approach, in which magnetic resonance imaging might be the first technique to suggest the diagnosis, which should be verified by an analysis of the *ROBO*3 gene. This is important to allow for adequate ocular follow up, apply supportive therapies to prevent the rapid progression of scoliosis, and lead to appropriate genetic counseling.

## 1. Introduction

Horizontal gaze palsy with progressive scoliosis (HGPPS; OMIM 607313) is a rare autosomal recessive disorder, first described by Dretakis and Kondoyannis [1] and with Crisfield’s first complete neurological description in 1974 [2], noting the absence of horizontal gaze in the patients examined and the development of severe and progressive scoliosis during childhood [3]. HGPPS is characterized by the congenital absence or severe restriction of horizontal gaze and progressive scoliosis that begins in early childhood. Currently, the treatment is through spinal surgery [4,5]. These patients appear to have few functional consequences, but results from standardized neuropsychological tests are currently unknown [6]. In 2004, this pathogenesis was linked to mutations on *ROBO3* genes in consanguineous families with autosomal recessive inheritance pattern of the disease [7]. Jen et al. [8] described that homozygous mutations occur on chromosome 11q23–25, which encodes a protein that shares homology with a member of the Roundabout (*ROBO*) gene family that controls neurite outgrowth, growth cone guidance, and axon fasciculation. Robo proteins are a subfamily of the immunoglobulin transmembrane receptor superfamily.

Robo3 is an axon guidance receptor exclusively expressed by Commissural (C-) neurons in the developing embryonic spinal cord [9]. C-neurons allow the coordination and integration of information coming from both sides of the body, which is essential for multiple functions such as binocular vision, sound localization, or integrated sensory-motor responses [10]. A normal functional *ROBO*3 gene is important in axon guidance activity and aids in the regulation of hind brain axon midline crossing [11]. Robo3 help in directing cell migration and specifies the lateral longitudinal pathways position. In addition, its interactions with other cell molecules support in the cytoskeleton assembly modification and growing axons regulation [7]. Forty-three different mutations located in diverse encoded protein domains have also been identified and are believed to decrease the axon-guidance receptor function [3]. Neuroimaging and neurophysiologic studies provide non-crossed motor and sensory pathways evidence in patients with HGPP [6]. Horizontal gaze palsy may be due to defects in the abducens nuclei (CN VI), which contain ipsilaterally and interneuronal motor neurons that project contralaterally, or supranuclear control regions, such as the paramedian pontine reticular formation (PPRF), which project to the abducens and oculomotor nuclei [7].

Published neuroimaging results have been normal in some cases, but not all [12]. The study conducted by Jen et al. [7] carried out high-resolution magnetic resonance imaging (MRI) in eight patients from four families and described that an abnormal flattening of the base of the pons and hypoplasia in the pontine tegmentum were evident in sagittal sections. Structural alterations in the inferior colliculus suggested a possible involvement of the abducens nuclei, the medial longitudinal fascicle, and the PPRF. The medulla frequently appeared butterfly-shaped, with anterior flattening and an unusual indentation in the midline. The abducens nerves were bilaterally visualized in the extraaxial space, and the orbital magnetic resonance images demonstrated normal extraocular muscle configuration and size, as well as the presence of apparently normal intraorbital motor nerves in the medial and lateral rectus muscles [7]. Despite these alterations found in magnetic resonance and electrophysiological studies, neurological functioning and sensorimotor integration seem to be intact in most patients [13]. There is little scientific evidence on the characteristics related to progressive scoliosis in patients with HGPPS. Most studies report that it should be treated surgically, and that begins in early childhood [4,5]. Mild or moderate scoliosis has also been described in some cases, but in many other cases, corrective surgery has been necessary due to the large angulation of the scoliotic curve [14].

The objective of this review is to group all HGPPS-related mutations described to date, to analyze the relationships between the most frequent characteristics of HGPPS—visual disturbances, the spine and the central nervous system, and to analyze the measuring instruments used to evaluate these patients.

## 2. Materials and Methods

This review was performed according to the Preferred Reporting Items for Systematic Reviews and Meta-Analyses (PRISMA) statement recommendations [15,16]. Search strategy used was “Horizontal gaze palsy OR binocular disorders OR gaze palsy) AND (scoliosis OR progressive scoliosis) AND (*ROBO*3 mutation OR mutation gene)”. Among the main outcomes, the severity of the syndrome and the impact on the patient’s life it had from an optometric and physiotherapeutic point of view were considered. To broadly approach the topic, a bibliographic search was made in the following electronic databases from January 1995 to October 2019: PubMed, Scopus, Web of Science, PEDRO, SPORT Discus, and CINAHL. The keywords used were horizontal gaze palsy, scoliosis, and children.

The titles and/or abstracts of studies retrieved using the search strategy and those from additional sources were independently screened by two review authors to identify studies that potentially met the inclusion criteria. The full text of these potentially eligible studies was retrieved and independently assessed for eligibility by two review team members. Any disagreement between them over the eligibility of studies was resolved through discussion with a third reviewer. A standardized, prepiloted form was used to extract data from the included studies for an assessment of study quality and evidence synthesis. The extracted information included study setting; study population and participant demographics and baseline characteristics; details of the intervention and control condition, measures and variables; study methodology; recruitment and study completion rates; outcomes and times of measurement; indicators of acceptability to users; suggested mechanisms of intervention action; information for assessment of the risk of bias. Missing data were requested from study authors by email with the corresponding author. The inclusion criteria were children with horizontal gaze palsy and progressive scoliosis with *ROBO*3 gene mutation. Among the main outcomes, the severity of the syndrome and the impact on the patient’s life it had from an optometric and physiotherapeutic point of view were considered. Finally, case reports, case series, randomized trials, and observational studies were the types of design that were included. The exclusion criteria were comments or letters to the editor, audiovisual documents (videos), non-English publications, and non-indexed publications. The authors designed the tables to extract the study data. This study selection process for this systematic review is described by a flow diagram (Figure 1).

The systematic review data information was extracted according to study characteristics and main outcome measures. The initial extracted data items comprised the following: (1) authors and publication year; (2) study design (case report or retrospective case series); (3) conflict of interest declaration (yes or no; if yes, which?); (4) subjects’ age (in years), and if the study had several subjects, the average was reported; (5) the percentage of females in the study; (6) relationship between the cases (if the study did not precisely report the family relationship, it was described as family); (7) ethnicity of the subjects; (8) parents’ relationship (if the study did not report the parents’ exact relationship, it was described as consanguineous); (9) best corrected visual acuity (BCVA) expressed on the Snellen scale; (10) visual axis alignment (the magnitude, eye, and orientation of the tropia was described when the study reported it); (11) horizontal gaze status (absence, palsy, restricted or convergence status); (12) vertical gaze status (limited or normal); (13) magnetic resonance imaging (MRI) findings with a description of brainstem and medulla characteristics; and (14) spine characteristics with scoliosis Cobb’s angle reported when available.

To determine the risk of bias for the individual studies, two independent and blinded reviewers with adequate reliability worked to create a summary chart (Appendix A) based on the Quality Assessment Tool for Case Series Studies from the National Heart, Lung, and Blood Institute [17]. In cases of non-agreement between the two reviewers, a third non-blinded reviewer broke the tie. Questions included in the tool were as follows. (1) Was the study question or objective clearly stated? (2) Was the study population clearly and fully described, including a case definition? (3) Were the cases consecutive? (4) Were the subjects comparable? (5) Was the intervention clearly described? (6) Were the outcome measures clearly defined, valid, reliable, and implemented consistently across all study participants? (7) Was the length of follow up adequate? (8) Were the statistical methods well described? (9) Were the results well described? This analysis does not lead to the elimination of any article. The articles with a risk of high bias were given a lower weight in the data synthesis section. We used the Scottish Intercollegiate Guidelines Network (SIGN) scale and use Methodology checklist 5: diagnostic studies in order to established and evidence level.

## 3. Results

A total of 25 case reports and case series published between 1998 and 2019 were included in this systematic review. Fourteen studies [3,4,8,12,14,18,19,20,21,22,23,24,25,26] were case series, and 11 [11,27,28,29,30,31,32,33,34,35,36] were case reports. Only nine studies [3,14,21,22,24,26,29,32,35] reported no conflict of interest, and the remaining studies [4,8,11,12,18,19,20,23,25,27,28,30,31,33,34,36] did not report conflicts of interest.

The age of the study subjects varied from 2 months to 55 years. The average age was 15.40 ± 15.50 years. Since all the studies were case reports or case series, the number of subjects was small. All case reports had a single case, while the case series reported between 2 and 10 cases. The average number of subjects in the case series was 3.8 ± 2.27 subjects. Fifty-four percent of the subjects included in the studies were female. Twelve [3,8,12,14,18,19,20,22,23,24,32,33] of the 25 studies reported the family relationships between subjects within studies. Most of the subjects were from Asia [3,8,11,19,22,23,25,29,31,32,33,35] followed by Europeans [12,14,21,24,25,34], and specifically mostly Eastern Europeans.

Only two studies [4,26] reported African subjects, and Bomfim et al. [30] studied subjects from South America. The relationship between parents was another point of study for this systematic review. Fifteen studies [3,4,8,11,12,14,18,20,22,24,25,27,30,33,35] described a consanguineous relationship between parents. Among the most common relationships were first cousins or children of first cousins. Eight studies [19,21,26,28,31,32,34,36] did not confirm the relationship between the parents of the subjects, and finally, the two remaining studies [23,29] did not report this information. Detailed study characteristics are reported in Table 1.

In the same way, the detailed outcome data extraction is reported in Table 2. Optometry and ophthalmology examinations included distance best corrected visual acuity (BCVA) expressed on the Snellen scale, visual axes alignment status expressed in prism diopters (∆) for both esotropia (ET) and exotropia (XT), and horizontal and vertical gaze status. In some cases [8,12,21,28], the horizontal gaze movement was accompanied by nystagmus.

Neurology and physiotherapy examinations described magnetic resonance imaging (MRI) findings with special attention to the characteristics and forms of the brainstem and medulla. The degree of scoliosis using the Cobb angle and its direction was also reported by most of the studies.

Axon guidance studies have suggested a model in which developing axons traverse a sequence of intermediate targets during development. Navigating these intermediate targets requires that developing axons respond to extracellular attractive and repulsive guidance cues, including members of the *netrin* and *slit* families, which are provided by specialized cells populations that reside along the axonal course. To date, every appropriately studied patient with complete or almost complete HGPPS had defined genetically homozygous or compound heterozygous mutation in the *ROBO*3 gene. Robo3 function loss has shown to prevent crossing at the ventral midline [9,38,39], indicating that it is required for commissure formation in the spinal cord and hindbrain.

This gene encompasses 28 exons and encodes a transmembrane receptor protein. It has a putative extracellular domain with five immunoglobulin (Ig1−5)-like loops and three fibronectin (FnIII1−3) type III motifs, a transmembrane segment (TM), and a cytoplasmic tail with three conserved signaling motifs: CC0, CC2, and CC3 (CC for conserved cytoplasmic) (Figure 2). Unlike other *ROBO3* family members, Robo3 lacks motif CC1. Zelina et al. [40] demonstrated that the key residues required for Slit1−3 binding in the Ig1 domain of Robo1 and Robo2 proteins have been substituted in the mammalian Robo*3* receptor and does not bind Slit1−3 with high affinity [38,41]; instead, they form a complex protein with the Netrin-receptor DCC (Deleted in Colorectal Cancer) through their cytoplasm domain [40]. All mammalian Robo3 receptors contain 10 conserved tyrosines on the cytoplasmic domains. Substitution of the conserved tyrosine residue (Y1019) in the CCO domain of Robo3 completely abolishes the Robo*3* phosphorylation induced by Netrin-1 [40]. HGPPS have identified multiple mutants in the *ROBO*3 gene [7]. The mutations determinates are highly diverse; most mutations are missense, nonsense, frameshift, and splice site mutations that affect multiple subdomains of the Robo*3* receptor supporting a complete Robo*3* function loss [42], as shown in Table 3.

HGPPS-related mutations occurred in all *ROBO*3 gene exons and exon–intron boundaries, which are mostly located on the extracellular protein [21]. *ROBO*3 domains or actions function need further research. Alternative splice forms of *Robo*3 [43] in the human brainstem [32], a phenotype–genotype correlation in HGPPS has not been obvious. In fact, it is unclear if *ROBO3* mutations alter ligand recognition, protein folding, or targeting and whether resultant changes in protein function might have a differential effect on developing nerve fiber tract decussating and/or on clinical expression such as scoliosis.

The risk of bias assessment for the studies was grouped into three outcome levels: low evidence level (between zero and three yeses), medium evidence level (between four and six yeses), and high evidence level (between seven and nine yeses). The following studies obtained a low evidence level: Rossi et al. (2004) [27], Amouri et al. (2009) [4], Bomfim et al. (2009) [30], and Ng et al. (2011) [11]. The following studies obtained a medium evidence level: Jen et al. (2002) [8], Pieh et al. (2002) [12], Lo et al. (2004) [19], Dos Santos et al. (2006) [28], Haller et al. (2008) [29], Abu-Amero et al. (2009) [37], Avadhani et al. (2010) [31], Abu-Amero et al. (2011) [32], Abu-Amero et al. (2011) [14], Jain et al. (2011) [23], Volk et al. (2011) [33], Bakbak and Kansu (2012) [24], Kurian et al. (2013) [25], Pina et al. (2014) [34], Bozdoğan et al. (2017) [35], Mendes Marques et al. (2017) [26], Lin et al. (2018) [36], and Rousan et al. (2019) [3]. Finally, the following studies obtained a high evidence level: Steffen et al. (1998) [18], Incecik et al. (2005) [20], and Chan et al. (2006) [21]. The level of evidence according SIGN scale was level 3. Non-analytic studies included case reports and case series.

## 4. Discussion

The selected studies present a total of 64 patients suffering from HGPPS, which is a congenital disease caused by an autosomal recessive disorder that is characterized by the restriction or absence of horizontal gaze and a progressive scoliosis that begins in early childhood [3,4,8,11,12,14,18,19,20,21,22,23,24,25,26,27,28,29,30,31,32,33,34,35,36]. Mutations in the *ROBO3* gene located on chromosome 11q23–25 are responsible for the development of horizontal gaze palsy and progressive scoliosis [2,4,32]. Jen et al. [8] set the HGPPS disease locus within a 30-cM region on chromosome 11q23–25. However, in the studies carried out by Lo et al. [19], the locus responsible for this pathology was redefined to a 9-cM region. There are more than 40 different mutations of the *ROBO3* gene published to date [32]. All of them affect the different dominions of the *ROBO3* gene. Nevertheless, to date, since no genotype–phenotype correlation has yet been elucidated in HGPPS, possibly because of intra-familial variability of the cardinal features [38], and whereas HGPPS with scoliosis has been described without detectable mutations in *ROBO3* gene [32], it is not possible to state that scoliosis is linked to *ROBO3* mutations [46]. In fact, it is unclear if *ROBO3* mutations alter ligand recognition, protein folding, or targeting and whether the resultant changes in protein function might have a differential effect on developing nerve fiber tract decussating and/or on clinical expression such as scoliosis.

The deficits in horizontal eye movement in HGPPS patients suggest that contralateral extraocular motor pathways are also affected, including contralateral inputs onto the abducens nucleus from the paramedian pontine reticular formation and projections from the abducens nucleus that target the contralateral oculomotor nucleus via the medial longitudinal fasciculus [42]. A HGPPS mouse study in which *ROBO*3 was conditionally knocked out in the hindbrain supports this analysis by reporting a reduction in contralateral projections at the level of the abducens nucleus and marginal connectivity between the abducens and contralateral oculomotor nucleus. However, the severe scoliosis that develops during childhood is less well understood and is thought to involve asynchronous muscle contractions, which underlie the breathing deficits in *ROBO*3 mutant mice [51,52] as well as defects in axial motor control [7].

According to Mendes Marques et al. [26], most families with a history of HGPPS belong to the Arab, Saudi, Turkish, Greek, Italian, American, and Chinese ethnicities. In our studies, most subjects were Asian [3,8,11,19,22,23,25,29,31,32,33,35], followed by Europeans (most from Eastern Europe) [12,14,21,24,25,34], and there were two studies that reported on African subjects [4,26] and one on South American subjects [30].

This syndrome has been described more frequently in individuals with inherited mutations in the homozygous state in consanguineous families [8]. In the present review, of the total subjects, 32 (48.48%) presented with inbreeding between parents [3,4,8,11,12,14,18,20,22,24,25,27,30,33,35]; the parents of 34 subjects (51.51%) were not consanguineous or that information was not reported [19,21,23,26,28,29,31,32,34,36].

Despite the different *ROBO*3 mutations and the various affected ethnicities, there are no significant differences in the clinical and imaging manifestations of patients with HGPPS [5]. This can be explained because in patients affected with HGPPS, mutations in the *ROBO*3 gene are identified, and this gene is essential for axons crossing the midline of the posterior brain and neuronal migration to the contralateral side during development of the nervous system [8,19]. The action of *ROBO*3 or its protein product might be inhibited by environmental or epigenetic factors in the developing brainstem; furthermore, a phenotype identical to HGPPS might be caused by abnormalities in *ROBO*3 splice variant expression. Moreover, although most reported *ROBO*3 mutations are equally distributed along the *ROBO*3 sequence, it would be interesting to determine whether specific mutation types are associated with a more disease phenotype and/or whether other disease genes for patients with horizontal gaze palsy with or without scoliosis who do not harbor mutations in *ROBO*3 are engaged.

The growing corticospinal and somatosensory axons cross the midline in the medulla to reach their objectives; therefore, these crossing axons form the basis of contralateral motor control and sensory input. Motor and sensory projections appear to have not crossed in patients with HGPP [28]. To evaluate these patients, brain magnetic resonance imaging evaluations had been used in all studies, except in one that did not mention this. These images show the characteristic congenital anatomical abnormalities of the brainstem that explain the clinical manifestations in these patients: signs of a divided protuberance, due to an abnormally developed medial longitudinal fascicle and abducens nuclei, a butterfly-shaped medulla, and the absence of the facial colliculus as a result of uncrossed corticospinal tracts [3,5].

All the selected studies found two common characteristics in the ophthalmic and optometric examinations: the restriction or absence of the horizontal gaze and the preserved vertical gaze. A horizontal gaze requires that the lateral rectus muscle of one eye, innervated by the abducens nerve, and the medial rectus muscle of the contralateral eye, innervated by the oculomotor nerve, work together. This coordinated activity is controlled by the abducens nucleus. This nucleus contains two populations of neurons: one that directly innervates the ipsilateral lateral rectus muscle and the other that consists of internuclear neurons that project through the medial longitudinal fascicle to the contralateral oculomotor nucleus to innervate the medial rectus muscle [27]. The abducens nucleus found at the bottom of the pontine tegmentum controls this activity. The paralysis in the horizontal gaze has been attributed to irregularly pronounced median longitudinal fascicular projections and anomalous innervations by the abducens supranuclear nerve [3,12,14,30,32,36]. Regarding the compensations for the horizontal gaze restriction with head movements, the research carried out by Haller et al. [29] concluded that in the future, functional magnetic resonance studies with a moving head could show if head movements that compensate for horizontal gaze paralysis activate the network of eye movements.

Progressive scoliosis is also considered a pathognomonic characteristic of this disorder. It is the most frequent reason for medical consultation in these patients because it produces an important functional limitation, lung involvement and pain, and it often requires surgery [25]. The different articles provided radiographs to show the severity of this dysfunction according to the Cobb angle. Different theories have been proposed as to why scoliosis occurs in these patients. One theory is the poor development of extrapyramidal projections in the reticular formation. The descending tracts of reticulospinal fibers in the reticular formation, together with the corticospinal tract, mediate control signals from the brain to the spinal cord to boost locomotion and regulate muscle tone [7]. Kurian et al. [25] attributed the abnormal control of axial tone to the involvement of the central tegmental tract combined with primary dysfunction in the musculoskeletal system as a result of mutations of the *ROBO*3 gene. Lin et al. [36] based their conclusions on the results of MR tractography, where the agenesis of afferent fibers within the inferior cerebellar peduncles and the pontocerebellar tracts can be seen. The publication of Ungaro et al. [53] concluded that there was no clear correlation between mutations in the *ROBO*3 gene and the cause of progressive scoliosis, nor if the pathophysiology was related to the nervous or musculoskeletal systems. An update recently published by Ungaro et al. [54] found 39 mutations.

### 4.1. Future Research Directions

The early clinical and neuroimaging diagnosis of HGPPS is fundamental for the prevention of the ocular and orthopedic problems that are associated with this pathology. Eye movement symptoms are noticed earlier than scoliosis [19]. Torticollis has been previously reported in several infants with HGPPS [21]. It requires a multidisciplinary diagnostic approach, in which MRI might be the first technique to suggest the diagnosis [3], which should be verified by the analysis of the *ROBO*3 gene. This is important to allow for adequate ocular follow up, apply supportive therapies to prevent rapid progression of scoliosis [28], and lead to appropriate genetic counseling [34].

### 4.2. Strengths and Limitations

To our knowledge, this systematic review is the first to compare studies to establish the relationships between the most frequent characteristics of HGPPS, visual disturbances, scoliosis, and the central nervous system, and analyze the materials used for these measurements.

Being considered a rare disease, the studies found that met the inclusion criteria were observations of a single case or series of cases with a maximum of 10 subjects. However, it must be said that there has been great progress since the relationship between the pathognomonic signs of this syndrome were first reported, and 30 years later, when it has been elucidated that the *ROBO3* gene and its mutations were responsible. This progress has been due to the identification of unique and interesting cases [32].

Regarding the limitations of this review, most of the studies demonstrated a moderate level of methodological quality. It should be noted that question 5, “Was the intervention clearly described?” and question 8, “Were the statistical methods wel-described? had “Not applicable” answer percentages of 84% (21/25) and 100% (25/25), respectively, because the studies did not perform any intervention, but only provided a description of the characteristics of the pathology.

## 5. Conclusions

According to the evidence level of the SIGN scale, the degree of diagnostic recommendation was D. The early diagnosis of HGPPS is important for the prevention of the ocular and orthopedic problems that are associated with this pathology. A multidisciplinary approach to this pathology is necessary for a correct diagnosis. Radiological studies, ophthalmological and optometric examinations, and genetic analyses must be carried out.

## Figures and Tables

**Figure 1 ijerph-17-04467-f001:**
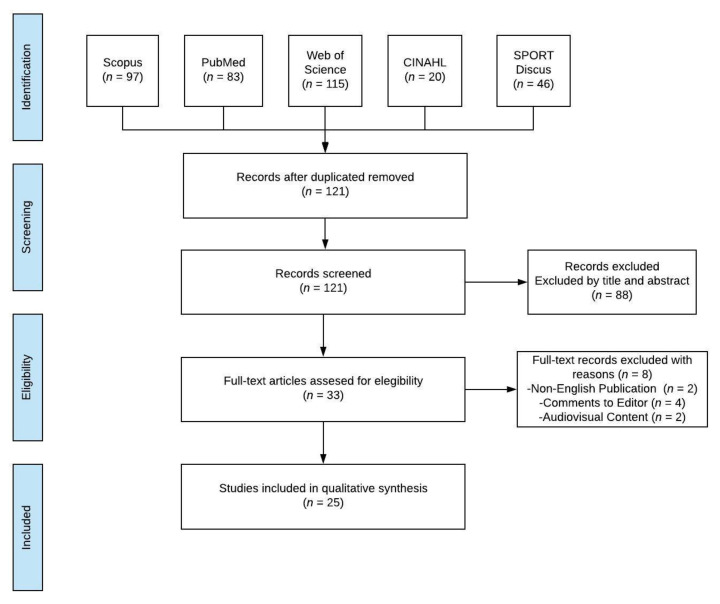
Preferred Reporting Items for Systematic Reviews and Meta-Analyses (PRISMA) flow diagram.

**Figure 2 ijerph-17-04467-f002:**
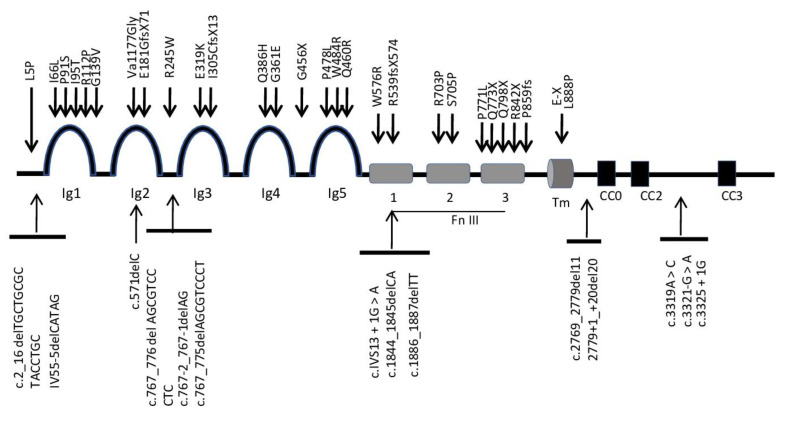
Roundabout (*ROBO*3) physical map domain with location of different mutations in patients with horizontal gaze palsy with progressive scoliosis (HGPPS).

**Table 1 ijerph-17-04467-t001:** Study characteristics.

Author (Year)	Design	Conflict	Age (Years)	Subjects	% Females	Cases Relationship	Ethnicity	Parents Relationship
Steffen et al. [18] (1998)	CS	NR	6.75	2	100	Sisters	NR	C
Jen et al. [8] (2002)	CS	NR	0.1–18	6	33	Family	Indian/Arabian	C
Pieh et al. [12] (2002)	CS	NR	32	2	0	Brothers	Italian	C
Lo et al. [19] (2004)	CS	NR	12	3	66.6	Siblings	Pakistani	NC
Rossi et al. [27] (2004)	CR	NR	13	1	100	NA	NR	C
Incecik et al. [20] (2005)	CS	NR	15	2	50	Siblings	NR	C
Chan et al. [21] (2006)	CS	No	3	2	50	None	Caucasian	NC
Dos Santos et al. [28] (2006)	CR	NR	2	1	0	NA	NR	NC
Haller et al. [29] (2008)	CR	No	14	1	0	NA	Kosovar	NR
Abu-Amero et al. [37] (2009)	CS	No	11.1	7	71	Brothers/Sisters	Saudi Arabian/Sudanese	C
Amouri et al. [4] (2009)	CS	NR	6–34	10	NR	NR	Tunisian	C
Bomfim et al. [30] (2009)	CR	NR	7	1	0	NA	Brazilian	C
Avadhani et al. [31] (2010)	CR	NR	14	1	0	NA	Indian	NC
Abu-Amero et al. [32] (2011)	CS	No	11.2	4	0	Brothers	Afghani	C
Abu-Amero et al. [14] (2011)	CR	No	8	1	0	Mother	Serbian	NC
Jain et al. [23] (2011)	CS	NR	10.5	2	100	Sisters	Indian	NR
Ng et al. [11] (2011)	CR	NR	55	1	0	NA	Indian	C
Volk et al. [33] (2011)	CR	NR	6.9	4	100	Sisters/None	Turkish/Saudi Arabian	C
Bakbak and Kansu [24] (2012)	CS	No	13	2	100	Sisters	Turkish	C
Kurian et al. [25] (2013)	CS	NR	6.75	2	100	None	Kosovar/Qatari	C
Pina et al. [34] (2014)	CR	NR	13	1	100	NA	Portuguese	NC
Bozdoğan et al. [35] (2017)	CR	No	22	1	100	None	Turkish	C
Mendes Marques et al. [26] (2017)	CS	No	12	2	50	NA	African	NC
Lin et al. [36] (2018)	CR	NR	55	1	100	None	NR	NC
Rousan et al. [3] (2019)	CS	No	12.8	6	66.6	Family	Jordanian	C

CR: Case Report; CS: Case Series; NR: Not reported; NA: Not applicable; C: Consanguineous; Non-Consanguineous.

**Table 2 ijerph-17-04467-t002:** Revised studies about horizontal gaze palsy with progressive scoliosis.

Author (Year)	BCVA	Tropias	Horizontal Gaze	Vertical Gaze	MRI	Spine
Steffen et al. [18] (1998)	20/30	NR	Absence	Normal	Normal	Cobb’s angle 45°
Jen et al. [8] (2002)	NR	ET (33%)	Absence	Normal	NR	Moderate scoliosis
Pieh et al. [12] (2002)	20/30	NR	Absence	Normal	BH and MO	NR
Lo et al. [19] (2004)	NR	Right ET	Palsy	Limited	BH and MO	Cobb’s angle 12°
Rossi et al. [27] (2004)	NR	NR	Absence	Normal	BH and MO	Severe scoliosis
Incecik et al. [20] (2005)	20/30	Normal	Restricted	Normal	NB and MM	Thoracolumbar scoliosis
Chan et al. [21] (2006)	20/40	15 ∆ ET	Absence	Normal	BH and MO	Profound scoliosis
Dos Santos et al. [28] (2006)	NR	Normal	Absence	Normal	BH and MO	Thoracolumbar scoliosis
Haller et al. [29] (2008)	NR	NR	Absence	Normal	BH and MO	Cobb´s angle 20°
Abu-Amero et al. [37] (2009)	NR	NR	Absence	Normal	BH	Left concave
Amouri et al. [4] (2009)	NR	NR	Absence	Normal	BH and MO	Thoracolumbar scoliosis
Bomfim et al. [30] (2009)	NR	NR	Absence	Normal	BH and MO	Mild thoracolumbar scoliosis
Avadhani et al. [31] (2010)	NR	NR	Absence	Normal	BH and MO	Cobb´s angle 32° and 63°
Abu-Amero et al. [32] (2011)	20/25	Mild ET	Absence	Normal	BH and MO	Kyphoscoliosis
Abu-Amero et al. [14] (2011)	NR	Mild left hypertropia	Absence	Normal	BH and MO	Concave 125° thoracolumbar kyphoscoliosis
Jain et al. [23] (2011)	20/32	25 ∆ ET	Absence	Normal	BH	Left kyphosis
Ng et al. [11] (2011)	NR	NR	Absence	Normal	BH and MO	NR
Volk et al. [33] (2011)	20/32	−10° XT	Absence	Normal	BH	Cobb’s angle 30°
Bakbak and Kansu [24] (2012)	20/35	NR	Absence	Normal	BH and MO	Moderate scoliosis
Kurian et al. [25] (2013)	NR	Left Tropia	Absence	Normal	BH and MO	Cobb’s angle 125°
Pina et al. [34] (2014)	20/80	Orthotropic	Palsy	NR	BH and AFC	Dorsal thoracolumbar scoliosis with torticollis over the left shoulder
Bozdoğan et al. [35] (2017)	20/32	Over 40 ∆ XT	Limited	Normal	BH and MO	Severe thoracic scoliosis
Mendes Marques et al. [26] (2017)	Normal	Orthophoria	Palsy	Normal	BH	Thoracolumbar scoliosis
Lin et al. [36] (2018)	Normal	NR	Palsy	Normal	BH and MO	Scoliosis thoracic and lumbar spine
Rousan et al. [3] (2019)	NR	Left hypertropia	Limited abduction	Normal	BH	Severe thoracic scoliosis

BCVA: best corrected visual acuity (Snellen scale); ∆: prism diopters; ET: esotropia; XT: exotropia; MRI: magnetic resonance imaging; NR: Not reported; BH: Brain hypoplasia; MO: Medulla oblongata or butterfly; NB: Normal brain; MM; AFC: Absence of facial colliculi.

**Table 3 ijerph-17-04467-t003:** Mutations in *ROBO*3 gene linked to HGPPS to date. Missense, nonsense, frameshift, and splice site mutations leading to premature stop codon and potentially truncated proteins.

Nucleotide Change	Exon	Amino Acid Change	Domain	Ethnicity	Reference
**c.14T > C**	1	L5P	Nter	Italian	Jen et al., [7] 2004
**c.2_16 delTGCTGCGCTACCTGC**	1			Saudi	Abu-Amero et al., [32] 2011
**IV55-5delCATAG**	2			Cape Verde	Mendes Marques et al., [26] 2017
**c.196A > C**	2	I66L	Ig1	Greek	Jen et al., [7] 2004
**c.271C > T**	2	P91S	Ig1	Sudanese	Abu-Amero et al., [37] 2009
**c.335G > C**	2	R112P	Ig1	Saudi	Abu-Amero et al., [37] 2009
**c.283T > C**	2	I95T	Ig1	Tunisian	Amouri et al., [4] 2009
**c.416G > T**	2	G139V	Ig1	Switzerland	Hackenberg et al., [44] 2016
**c.571delC**	2	frameshift	Ig2	Saudi	Abu-Amero et al., [37] 2009
**c.530dlT**	3	Va1177Glyfs * 45 frameshift	Ig2	Jordanian	Rousan et al., [3] 2019
**c.541dup**	3	E181GfsX71	Ig2	Kosovar	Kurian et al., [25] 2013
**c.733C > T**	4	R245W	Ig2–3	Irish/English	Chan et al., [21] 2006
**c.733C > T**	4	R245W	Ig2–3	Tunisian	Amouri et al., [4] 2009, Chan 2006 [21]
**c.767_776delAGCGTCCCTC**	5	c.767_776delAGCGTCCCTC	Ig2–3	Portuguese	Pina et al. [34] 2014
**c.767-2_767-1delAG**	5	c.767-2_767-1delAG	Ig2–3	Portuguese	Pina et al. [34] 2014
**c.767_775delAGCGTCCCT**	5	c.767_775delAGCGTCCCT	Ig2–3	Cape Verde	Mendes Marques et al., [26] 2017
**c.955G > A**	6	E319K	Ig3	Greek	Jen et al., [7] 2004
**c.913delAinsTGC**	6	I305CfsX13	Ig3	Caucasian/Turkish	Volk et al., [33] 2011
**c.1082G > A**	7	G361E	Ig4	Indian	Jen et al., [7] 2004
**c.1158G > C**	7	Q386H	Ig4	Spanish	Fernández-Vega Cueto et al., [45] 2016
**c.1366G > T**	9	G456X	Ig4–5	Turkish	Jen et al., [7] 2004
**c.1379A > G**	9	Q460R	Ig5	Saudi	Abu-Amero et al., [37] 2009
**c.1450T > C**	9	W484R	Ig5	Tunisian	Amouri et al., [4] 2009
**c.1433C > T**	9	P478L	Ig5	Italian	Ungaro et al., [46] 2018
**c.1618delG**	10	R539fsX574	Fn III 1	Tunisian	Amouri et al., [4] 2009
**c.1726T > C**	11	W576R	Fn III 1	Saudi	Abu-Amero et al., [37] 2009
**c.1886_1887delTT**	12	frameshift	Fn III 1	Irish/German	Chan et al., [21] 2006
**c.1844_1845delCA**	12	frameshift	Fn III 1	Irish/German	Chan et al., [21] 2006
**c.IVS13 + 1G > A**	13	frameshift	Fn III 2	Saudi	Jen et al., [7] 2004
**c.2108G > C**	14	R703P	Fn III 2	Turkish	Jen et al., [7] 2004
**c.2113T > C**	14	S705P	Fn III 2	Saudi	Jen et al., [7] 2004
**c.2310 + 1C**	15	frameshift	Fn III 3	Pakistani	Jen et al., [7] 2004
**c.2317C > T**	15	Q773X	Fn III 3	Irish/English	Chan et al., [21] 2006
**c.2312C > T**	15	P771L	Fn III 3	Saudi	Khan et al., [47] 2008
**c.2392C > T**	15	Q798X	Fn III 3	Japanese	Yamada et al., [48] 2015
**c.2576del**	16	P859fs	Fn III 3	Austrian	Arlt et al., [49] 2015
**c.2524C > T**	16	R842X	Fn III 3	Turkish	Bozdoğan et al., [35] 2017
**G > T ***	17	^E-X	TM	Indian	Ng et al., [11] 2011
**c.2663T > C**	17	L888P	TM	Saudi	Khan and Abu-Amero, [50] 2014
**c.2769_2779del11**	17	Splicing defect + frameshift	TM-CC0	Caucasian/Turkish	Volk et al., [33] 2011
**2779+1_+20del20**	17	Splicing defect + frameshift	TM-CC0	Caucasian/Turkish	Volk et al., [33] 2011
**c.3319A > C**	22	skip + frameshift	CC2-CC3	Caucasian/Turkish	Volk et al., [33] 2011
**c.3321-G > A**	22	int 22 Splice site mut	CC2-CC3	Italian	Ungaro et al., [46] 2018
**c.3325 + 1G**	23	frameshift	CC2-CC	Saudi	Jen et al., [7] 2004
**c.3742C > T**	25	(Arg1248 *) nonsense	CC3	Jordanian	Rousan et al., [3] 2019

* The nucleotide number was not mentioned by the authors. ^ The codon number was not mentioned by the authors.

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
