# Peer review of "Mutation in ROBO3 Gene in Patients with Horizontal Gaze Palsy with Progressive Scoliosis Syndrome: A Systematic Review"

_ijerph, 2020, doi:10.3390/ijerph17124467_

Round 1
Reviewer 1 Report
Some comments are suggested:
- The use of MeSH descriptors is recommended as keywords.
- All abbreviations must be clarified
- The methodology must specify how the study selection was carried out.
- In the inclusion criteria, it must be divided between the participants, the main objectives, the types of design and the intervention.
- Line 107 to 109 repeats "Two review authors independently extracted data, and discrepancies were identified and resolved through discussion (with a third author where necessary)".
- In line 111 to 113, inclusion and exclusion criteria are indicated again. All should be organized together as indicated above
- What difference is there between the extraction of data from the articles indicated from line 103 and the one that appears from line 115?
- It should be much clearer what designs have been included in the criteria, since the results refer only to case reports and case series. Explain why other types of designs have not been included.
- When means are provided, it would be interesting to include the mean standard deviation of the included studies
- It would be interesting to include the bias graph.
Reviewer 2 Report
The authors performed an extensive review of the available medical literature in patients with Horizontal Gaze palsy with progressive scoliosis syndrome. The showed the correlation of the Robo3 gene mutations. This information would lead to earlier recognition of this disease entity and help with ophthalmologic, orthopedic management, and appropriate and timely genetic counseling.
Author Response
Dear reviewer,
Thank you for the comments made on our manuscript, it is very important to us.
Regards, Veronica Perez-Cabezas and co-authors.
Reviewer 3 Report
see attach

This manuscript is a resubmission of an earlier submission. The following is a list of the peer review reports and author responses from that submission.
Round 1
Reviewer 1 Report
In the current manuscript the authors have attempted to review a large number of case studies on the HGPPS disease. The efforts are commendable and this is something the authors wanted the readers to appreciate through their methodology as well as the table and a figure. The in-depth explanation of the method section drives the expectations high. However, the results section is weak and can be put in a simple table. My expectations in the results section were on the findings on genes, specific mutations, molecular pathways etc. as indicated in the topic. Nowhere it mention about specific mutation, its position in the genome, type of mutation involved, protein or amino acid changes due to specific mutation, etc. especially in the post genome era. The discussion section again does not do justice to the extensive review-of-literature and simultaneously fails to summarize or conclude the role of specific mutation or broadly ROBO3 gene/protein.
Specific comments.
Title: The gene name should follow HUGO gene nomenclature.
In most cases, mutations are inherited in a homozygous state in consanguineous families [7] Which mutation? The reference [7] appears inappropriate for this statement because the conclusion of this reference by Abu-Amero et al, 2011, states, “HGPPS can be caused by abnormalities in ROBO3 splice variant expression, by mutations of a gene other than ROBO3, or by some environmental or epigenetic factor(s) inhibiting the action of ROBO3 or its protein product in the developing brainstem” which is also clear form the title itself, “… without ROBO3 mutations….”.
Additionally, this statement also is misleading because the author mention that the mutations are ‘inherited in a homozygous’ state, which I believe should be “inherited the same mutation from both parents”
Line 52: …function of this receptor 3… explain ??
identify ROBO3 gene mutations in patients with HGPPS??
Table 1 although seems worthy but does not contribute to the advancement to the knowledge on the overall topic. Can be supplementary.
The manuscript needs a major alteration so that it is re-written in a typical review format rather than in a research article format. The main reason for this recommendation is because the entire results section does not add much to the core subject the authors have endeavored to highlight and resolve. Some part from the results although is informative and therefore by restructuring the manuscript under different heading e.g., population specificity, details on specific genes involved, of role of ROBO3 and additional gene???, association of ROBO3 gene with specific or multiple phenotype/s, etc..
By providing more details on the phenotypes, as indicated in different Tables, the focus of the review appears to deviate away from the “Mutation in ROBO3 gene”??
Reviewer 2 Report
The systematic review has been followed through the PRISMA recommendations and the selection of articles has been met with quality criteria.
It would have been interesting if the authors had included the systematic review in PROSPERO
Some comments are suggested:
- Abbreviations should not appear in the abstract, except those referring to databases.
- The use of MeSH descriptors is recommended as keywords. In the case of gaze palsy, it is not a descriptor per se, but a complementary concept (Gaze Palsy, Familial Horizontal, with Progressive Scoliosis [Supplementary Concept]), so it must be taken into account in this way for use in the search.
- Paragraph 47-57 could be improved in terms of wording and punctuation marks to improve clarity.
- The risk of bias assessment must also be expressed through the chart established by the Cochrane
- You must specify which checklist was used for critical reading of the articles. In addition, it is recommended to include an analysis of the level of evidence, for example according to the SIGN scale.
- In the conclusions it would be interesting to include a degree of recommendation for the diagnosis according to the results obtained in the review.
Thank you
Reviewer 3 Report
The authors present an interesting article discussing the mutation in ROBO3 gene in patients with Horizontal Gaze Palsy with Progressive Scoliosis syndrome. While the authors did extensive review of the literature according to the PRISMA guidelines, it is hard to link the results section to the discussion and the conclusion. There is no inference in the results.
Round 2
Reviewer 1 Report
In order to address the previous queries and concerns by the reviewers the authors have now included some of the genetic mutations that are directly related to HGPPS disease. This was necessary especially when the title indicates ‘Mutation in ROBO3…’.
The author still needs to denote correct references of some of the statements especially articles from the same authors are misrepresented in the references. For example, the statement “… that homozygous mutations occur on chromosome 11q23–25” refer to another reference by the same group.
The author needs to utilize the empty space, especially in the two figures as it adds un-necessary extra pages (not sure if this is something the journal takes care while editing??).
